# Proactive Operations Management: Staff Allocation with Competence Maintenance Constraints

**Eryk Szwarc [1], Grzegorz Bocewicz [1,*], Paulina Golińska-Dawson [2] and Zbigniew Banaszak [1]**

[1] Faculty of Electronics and Computer Science, Koszalin University of Technology, ul. Śniadeckich 2, 75-453 Koszalin, Poland

[2] Faculty of Engineering Management, Poznan University of Technology, ul. Jacka Rychlewskiego 2, 60-965 Poznań, Poland

\* Correspondence: grzegorz.bocewicz@tu.koszalin.pl

**Abstract:** Highly qualified staff are the key to successful operations management in any organization. In this paper, the emphasis is put on the problem of planning the rotational assignment of work tasks to a multi-skilled staff to guarantee maintaining their competencies at the required level. The aim of this study is to propose a novel declarative model for proactive planning of staff allocation whilst taking into account the forgetting effect. Sufficient conditions are proposed that allow for the cyclical rotation of employees between different tasks in order to keep their competencies at a constant level. The numerical experiments prove that the presented approach allows for finding a trade-off between a robustness to absenteeism and maintaining staff competency levels. The proposed method is suitable for human resource-related decision making in an interactive mode.

**Keywords:** proactive planning; workers assignment; competence maintenance; robustness

## 1. Introduction

Human resources are crucial for successful and sustainable operations management in knowledge-intensive sectors, e.g., education [1,2]. To maintain a competitive position in fast-changing conditions, organizations must search for ways to maintain staff competence while avoiding unnecessary training costs. In this paper, we use the education sector as an example of an area where multi-skilled employees perform various tasks over different periods of time [3]. The problem of assigning courses to be taught in academic organizations (polytechnics and universities) combines elements such as staffing (perceived in the context of assigning teachers to courses, i.e., the teacher assignment problem (TAP)), and their scheduling (course timetabling and scheduling) [4,5]. The planning of courses determines which teacher will conduct each course whilst taking into account factors such as the competencies and preferences of teachers and their working hours. Additionally, the scheduling of courses (designing a feasible timetable) in so-called time windows (usually weekly), is important. An acceptable schedule should not contain conflicts or overlays leading to situations where a teacher is scheduled to conducts two meetings at the same time, rooms are double-booking, or where more than one course for the same group of students is scheduled in a given time window. Furthermore, additional factors must be included such as individual user preferences (e.g., only available until 4:00 p.m.), respecting daily limits of hours (e.g., no more than 6 h), etc. In practice, these problems are usually solved sequentially, i.e., first staffing and then scheduling [6]. The staffing and scheduling of academic staff for courses is subject to limitations such as daily and weekly limits of working hours, even workloads, availability of qualified staff, and room availability.

In this paper, we focus on the availability of multi-skilled academic staff and maintaining their competencies through rotation. Competencies are defined here as a set of knowledge, experience, and skills that are necessary to conduct a predefined set of

courses. The competence structure of a team is created by the individual competencies of each member of the team [7].

The nominal plan of courses can be disrupted by employee absenteeism (sick leave, accidents, etc.), a loss of qualifications, termination of employment, etc. As a result, the employer may have to hire replacement teachers on an ad hoc basis. To protect the organization from the effects of these disruptions, decision support methods and IT solutions can be used. These solutions support decision-makers in planning so-called "live competence structures", which enable the implementation of planned activities despite emerging disruptions [8].

Previous research has focused on the construction of competence structures that guarantee the implementation of courses in static [9] and dynamic conditions, taking into account the possibility of the occurrence of disruptions, for example, teacher absenteeism [10]. The assumptions of the developed models provide for the acquisition of some competencies without losing others. Competencies, if they are not used, disappear over time. This is often observed in practice resulting in the forgetting effect and leads to a constant loss of skills and knowledge [11]. It can be caused, inter alia, by:

- A break in the use of a specific skill or knowledge,
- Aging competencies,
- Biological ageing, diseases, and accidents.

The loss of competencies due to the forgetting effect can cause changes in competence structures and, therefore, reduce the university's robustness to teacher absenteeism. As a result, this can lead to the weakening of teacher competences and additional costs for staff training and hiring. Some of these challenges are related to the rational use of experienced staff whose are competent in teaching several different courses. The schedules maintained by these experienced staff should guarantee a periodic rotation of the courses taught, thus avoiding the forgetting effect.

In this paper, the emphasis is placed on the problem of planning the cyclical assignment of courses to a multi-skilled staff to guarantee the implementation of courses without losing the competences already possessed by a given employee. This paper proposes a novel declarative model for proactive planning of staff rotational allocation whilst taking into account the forgetting effect.

Previous studies have not proposed comprehensive models for balancing available multi-skilled employees with the requirements resulting from workspan. Furthermore, there is a lack of proactive planning methods for the assignment and rotation of a team of employees (through synthesis of a suitable competence structure) when ad hoc disruptions occur. There is a need for analytical models in the area of human resources which enable efficient rotation to maintain staff competencies at a required level. A comprehensive literature review and a detailed definition of the research gaps are provided in Section 2.

The decision-making dilemma results from the mutual contradiction between the willingness to maximize the robustness of a competence structure and the need to maintain a constant level of team competencies. This work expands on previous studies investigating the allocation of human resources [10,12,13], by including of the forgetting effect [14].

The novelty of this study results from:

- Proposing of generic model for proactive allocation of a multi-skilled workforce whilst taking into account the forgetting effect,
- Defining sufficient conditions for cyclical relocations of employees to maintain their competence at a constant level,
- Developing a method for selecting the competences of team members aimed at a trade-off between the assumed robustness to absenteeism and maintenance of the competencies of team members through cyclical rotation of positions.

The remainder of the paper is organized as follows. In Section 2 we discuss studies related to work allocation and staff scheduling taking into account the effect of learning and forgetting. In Section 3 an illustrative example introduces a generic model of teacher assignment planning. In Section 4 we describe the reference model of teaching staff assignment that is aimed at maintaining their competencies and robustness. Afterward, Section 5 presents a university case study. Section 6 presents experiments conducted to analyze the performance of the proposed approach. Section 7 concludes the paper and presents future work.

## 2. Literature Review

The changing conditions of the organization's operation, in particular those related to the pandemic period, underlined the value of managers' competencies related to the risk assessment of human resources availability [15]. To achieve its business goals, an organization must carefully balance the resources it already has and those it needs to complete given orders. A trend observed in recent years shows that particular attention should be given to the employees' competence structures, which determine whether or not a company is capable of completing orders.

The competence structure covers a set of competencies necessary to perform specific tasks in a company, conditioning the adaptation of personal skills and accumulated knowledge to achieve its goals. The term skills are usually understood as the ability of a worker to perform certain tasks well [16,17]. The related literature abounds in studies on methods for supporting decisions on assessing employee competencies, identifying competence gaps, prototyping competence changes, planning the allocation of employees to operations, etc.

### 2.1. Work Allocation and Staff Scheduling

A fast-growing body of literature focuses on topics regarding personnel scheduling issues [18] such as crew scheduling, shift scheduling, and personnel assignments [19,20]. These studies fall within the area of work allocation while specifying when and for how long those tasks should be performed [12]. The interlacing problems of scheduling and workforce assignment involve the allocation of employees with different competencies, to activities carried out within the given time intervals. Both problems are combinatorially NP-hard [21]. For this reason, in situations where accurate solutions are required, methods such as mixed integer linear programming [13,22], constraint logic programming [23], and Hungarian methods [24] are used, but their implementation is limited in practice to small scale problems. For bigger scale problems, the application of AI methods is required, especially those based on genetic algorithms [25], as well as stochastic and fuzzy set-based techniques [26,27].

The dynamic nature of the environment surrounding work allocation (job cancellation, delays, sickness, regulation changes, etc.) forces such a system to be very responsive and flexible. In such circumstances decision-makers need to be able to predict disruptions such as employee absences (sick leave, accidents, maternity leave, etc.), loss of qualifications (associated with the occurrence of the forgetting effect or the need to update certificates, such as driving licenses, electrician licenses, etc.), changes in the number of activities, tasks or jobs (caused by the addition or removal of a currently serviced order), loss of employees (employ walkouts), etc. A review of the literature shows that issues related to protecting organizations against the effects of such disruptions are rarely discussed. The techniques used to address these issues assume that an organization should have redundant human resources (including multi-skilled ones). Unfortunately, there are still no solutions in this area for supporting decision-makers in planning competence structures that can guarantee the completion of planned project portfolios in dynamically changing project implementation conditions.

In this context, an approach based on a proactive allocation of personnel provides quite promising solutions. This applies the concept of a robustness of a competence structure [8] with the predefined characteristics of the skills of the employees and features of an organization (e.g., project portfolio). A competence structure, or a competencies (skills) matrix [28], is used as a tool for specification and visualization of staff skills. In previous research [7,10], it has been shown that such matrices, in addition to the description of the staff competencies also can be used to identify the needs for improving qualifications and increasing the productivity of the organization. They can be successfully used to assess the robustness of employee teams to disruptions caused by absenteeism or the appearance of unexpected high priority jobs. A quantitative measure of the robustness of a competence structure determines whether an organization's personnel are able to take on additional responsibilities (substitutions) when a certain number of employees are absent or when additional tasks are added to the current schedule. The hiring of multiple professional staff to increase robustness to this type of disruption is limited by the occurrence of the forgetting effect [29], which forces the rotation of staff.

### 2.2. Competence Maintenance

The forgetting effect derives from the learning curve model [30]. It has been used in the automotive [31], machine [32], electronic [33], and construction [34] industries. Jaber and Bonney [35] developed the learning forgetting curve model (LFCM). This model assumes that the rate of forgetting depends on the speed of learning, the minimum production break followed by complete forgetting, and the amount of accumulated experience that the operator has at the beginning of the break. The LFCM's comparative analysis with other models such as RC (recency) and PID (power integration diffusion) can be found in the works of Hoedt et al. [29] and Jaber and Sikström [36].

Models of learning and forgetting have found their application in work assignment [37,38]. They assume that breaks in the performance of work (depending on the adopted forgetting coefficient) prolong its duration. It should be noted that, in the teacher assignment problem (TAP), breaks in commissioning specific courses do not prolong their duration. Instead, discontinuity (irregularity) in conducting particular courses causes a partial or complete loss of competence, which can affect the robustness of the competence structure to selected disruptions, as described in previous studies [8,10].

The literature indicates that to overcome the forgetting effect, activities should be reperformed (a form of repetition or consolidation of knowledge and skills), which allows one to maintain competencies at the required level. This statement coincides with the well-known method of job rotation, which can be summarized as the structured interchange of workers between different jobs at certain time intervals. Many studies have been devoted to research in this area [39–43]. This is due to observations that indicate the benefits of job rotation and include, among other things, enhanced productivity following acquired skills and knowledge [44], as well as development of employee qualifications by accumulating new experiences [45]. Unfortunately, due to the aforementioned forgetting effect, extending the scope of competencies requires a systematic refreshment (certification). In practice, this creates the necessity, as in the industrial environment, to introduce appropriate maintenance mechanisms [46,47]. Relating this association to the academic community, it is easy to notice that maintaining the competencies enabling a lecturer to conduct several different courses requires their periodic repetition (updating). This is connected with the necessity of setting schedules (timetables) with a cyclical rotation of conducted courses. Unfortunately, the research conducted in the field of maintaining human resources, and in particular maintaining acquired competencies, is in the initial stages of development.

### 2.3. Research Gaps

The following important research gaps can be summarized from our literature review:

- There are no comprehensive models which would enable the study to balance the available human resources (in particular, multi-skilled staff members) with the requirements resulting from particular orders,
- There is a lack of methods for the proactive management of human resources, in particular proactive planning of assignment and rotation of employee tasks, i.e., enabling the formation of employee teams (through synthesis of the suitable competence structure) guaranteeing the timely execution of orders in situations related to ad hoc occurring disruptions,
- There is a lack of analytical models for maintaining multi-skilled human resources that would allow for the development of job (task) rotation methods ensuring the maintenance of staff competencies at a constant level.

## 3. Maintaining Staff Competencies

Maintaining the required competence of the team depends on competencies possessed by individual members. Teams with multi-skilled employees are characterized by lower sensitivity to the occurrence of employee absenteeism, and thus less risk associated with not completing a task. Increasing the team's robustness by hiring employees with many competencies makes it difficult to plan a feasible rotation schedule which guarantees the maintenance of all competencies. The problem of determining the competence structure (matrix) with a given level of robustness to absenteeism might not be solved in an acceptable way. Consequently, it is necessary to set sufficient conditions to guarantee the existence of a nonempty set of acceptable solutions. Such conditions must take into account the parameters characterizing individual team members, including the sets of competencies and the periods of time which are necessary to maintain them (renewal, certification, upgrade etc.). Compliance with these conditions allows cyclical teacher assignment to maintain the necessary competencies. To do so, we use two basic indicators for the assessment of the competence structure, namely: the degree of robustness and competencies lifetime.

### 3.1. Competence Structure Assessment Indicators

In order to define the factors determining the assessment of a competence structure, let us consider the example in which six academic teachers $P = (P_1, \ldots, P_6)$ perform academic curriculum $Q$ consisting of eight courses $Z = (Z_1, \ldots, Z_8)$. The teacher competencies are presented in the competence structure $G$ (Table 1). The value of 1 means that an employee is competent to perform a course, and the value of 0 means the opposite situation. For example, teacher $P_1$ can provide courses $Z_2, Z_6, Z_7, Z_8$, and cannot provide courses $Z_1, Z_3, Z_4, Z_5$.

**Table 1.** Competence structure $G$.

| $G$ | $Z_1$ | $Z_2$ | $Z_3$ | $Z_4$ | $Z_5$ | $Z_6$ | $Z_7$ | $Z_8$ |
|---|---|---|---|---|---|---|---|---|
| $P_1$ | 0 | 0 | 0 | 0 | 0 | 1 | 1 | 1 |
| $P_2$ | 0 | 0 | 1 | 1 | 0 | 0 | 1 | 0 |
| $P_3$ | 0 | 1 | 0 | 0 | 1 | 0 | 0 | 0 |
| $P_4$ | 0 | 0 | 1 | 1 | 1 | 0 | 0 | 0 |
| $P_5$ | 1 | 0 | 0 | 0 | 0 | 1 | 0 | 1 |
| $P_6$ | 1 | 1 | 0 | 0 | 0 | 0 | 0 | 0 |

Curriculum $Q$ is repeated in subsequent periods $T_l$ (semesters or years) with assumptions:

- each course $Z_i$ can be assigned to only one teacher $P_k$ in $T_l$,
- each teacher $P_k$ provide at least one course $Z_i$ in $T_l$.

In Table 2 is given the course assignment $X$. The value of 1 means the assignment of the courses to a specific teacher, and the value of 0 no assignment. For example, teacher $P_1$ is assigned to course $Z_8$, teacher $P_2$ is assigned to courses $Z_4$ and $Z_7$, etc.

**Table 2.** Course assignment $X$.

| $G$ | $Z_1$ | $Z_2$ | $Z_3$ | $Z_4$ | $Z_5$ | $Z_6$ | $Z_7$ | $Z_8$ |
|-----|-----|-----|-----|-----|-----|-----|-----|-----|
| $P_1$ | 0 | 0 | 0 | 0 | 0 | 0 | 0 | 1 |
| $P_2$ | 0 | 0 | 0 | 1 | 0 | 0 | 1 | 0 |
| $P_3$ | 0 | 0 | 0 | 0 | 1 | 0 | 0 | 0 |
| $P_4$ | 0 | 0 | 1 | 0 | 0 | 0 | 0 | 0 |
| $P_5$ | 1 | 0 | 0 | 0 | 0 | 1 | 0 | 0 |
| $P_6$ | 0 | 1 | 0 | 0 | 0 | 0 | 0 | 0 |

It is assumed that assignment $X$ is repeated in subsequent periods $T_1, T_2, T_3$ (Figure 1), and the disruption caused by a single absenteeism of a teaching staff member in period $T_l$.

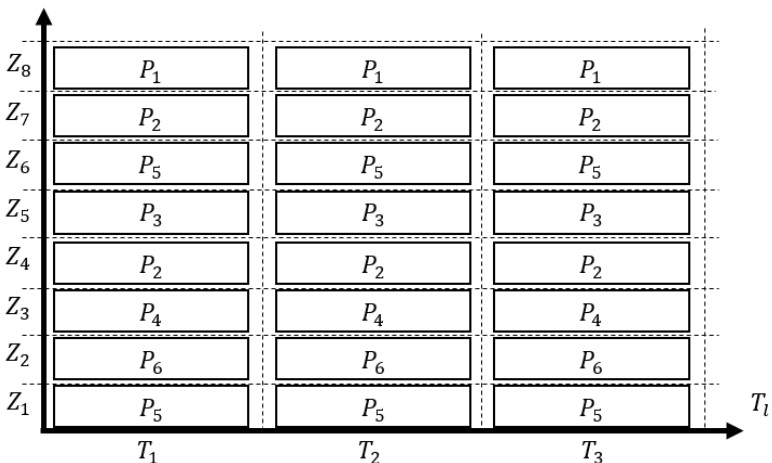

**Figure 1.** Schedule of assignment $X$ from Tab 2 repeated in $T_1, T_2, T_3$.

### 3.1.1. Robustness to Disruption (Absenteeism)

The occurrence of such a disruptions forces replacement for courses assigned to the absent teachers. In general, absences are usually unexpected (unplanned). This means that it is difficult to predict which teacher will be absent and when (in which period). For this reason, it is necessary to assume the absence of each employee in each period. For example:

- $P_1$ absenteeism in $T_1$ and requires replacement for the course $Z_8$,
- $P_3$ absenteeism in $T_2$ and requires replacement for the course $Z_5$,
- $P_5$ absenteeism in $T_3$ and requires replacement for the courses $Z_1$ and $Z_6$.

Whether replacement is possible depends on the competence structure (see Table 1), inter alia:

- in $P_1$ absenteeism scenario, the course $Z_8$ may be carried out by $P_5$,
- in $P_3$ absenteeism scenario, the course $Z_5$ may be carried out by the $P_4$,
- in $P_5$ absenteeism scenario, the course $Z_1$ may be carried out by the $P_6$, and the course $Z_6$ may be carried out by the $P_1$.

In reference to previous research [7,10], the competence structure's ability to cope with disruption can be assessed by measuring the competence structure robustness (CSR),

i.e., ratio of the number of disruption scenarios $(LP)$ for which the competence structure $G$ guarantees replacement for absent $k$-th teacher, to all possible distribution scenarios $(U)$:

$$R = \frac{LP}{U} \tag{1}$$

In the discussed example there are 18 possible scenarios of single teacher absence $(U = 18)$, and for all of them, the replacement by another competent teacher exists $(LP = 18)$. According to (1) it means that CSR is $R = 1$.

The presented example assumes that the competence structure is not changing in subsequent periods $T_l$. In consequence, an assignment $X$ (see Table 2) can be fixed (repeated in $T_1, T_2, T_3$—according to Figure 1). This leads to a situation where some competencies are not used (e.g., teacher $P_1$ does not use the competencies for courses $Z_6, Z_7$; teacher $P_2$ does not use the competencies for course $Z_3$; etc.). In practice, many teachers who do not teach a specific course, lose the knowledge and skills related to it after some period of time. In other words, teachers experience the so-called forgetting effect.

### 3.1.2. The Forgetting Effect

Let us introduce the parameter "competence lifetime" denoted by $cl_{k,i}$, which means the time after the teacher $P_k$ loses (forgets) competence to perform course $Z_i$ and is described by the forgetting function $F_{k,i}(l)$ according to [48] (see Figure 2):

$$F_{k,i}(l) = l^{-f_{k,i}}, \tag{2}$$

where $l$ is a number of periods $T_l$, $f_{k,i}$ is the forgetting curve slope for teacher $P_k$ and course $Z_i$: $f_{k,l} \in \{0, 1\}$.

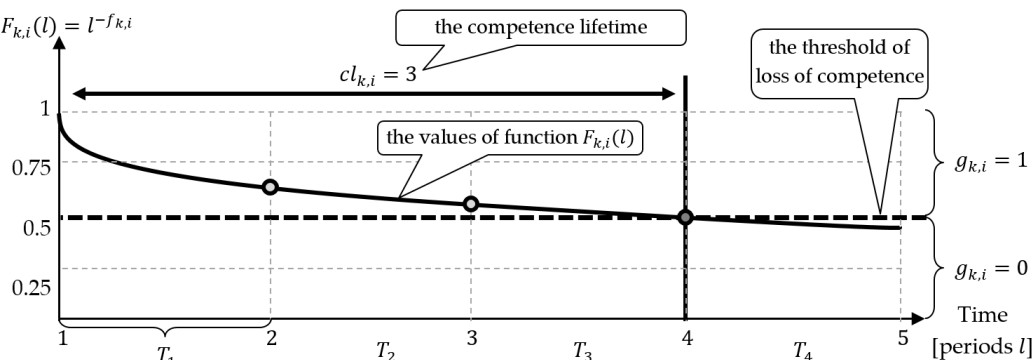

**Figure 2.** An example of the forgetting function $F_{k,i}(l)$ with the competence lifetime parameter $cl_{k,i}$.

The level of competencies in the period $T_l$ is represented by matrix $\Lambda^l$:

$$\Lambda^l = \left[\lambda_{k,i}^l\right]_{k=1,\dots,m;\ i=1,\dots,n'} \tag{3}$$

where $\lambda_{k,i}^l$ is the competence level of the teacher $P_k$ for the $Z_i$ course: $\forall_{g_{k,i}=1} \lambda_{k,i}^l \in \{0, \dots, cl_{k,i}\}$ and $\forall_{g_{k,i}=0} \lambda_{k,i}^l = \emptyset$ (lack of a competence).

The competencies can be "refreshed" by conducting courses related to them (after execution: $\lambda_{k,i}^l = cl_{k,i}$) and in general cases should be refreshed more frequently than it is determined by the lifetime $cl_{k,i}$. For example, let us assume $cl_{k,i} = 2$ for each $k$-th teacher and each $i$-th course. The schedule from Figure 1 will lead to the loss of some competencies (see Figure 3a), e.g.,:

- $P_1$ will lose competences in $Z_6$ and $Z_7$ in the $T_2$ period,
- $P_2$ will lose competence in $Z_3$ in the $T_2$ period.

Furthermore, of the 18 possible scenarios of a single teacher absence ($U = 18$), the possibility of replacement by another competent teacher only exists in 12 ($LP = 12$). According to (1) it means that the CSR is $R = 0.66$. In other words, lost competencies affect the CSR level, and threaten the ability to carry out courses. It should be noted that by rotating the assignment in subsequent periods $T_l$, competencies can be refreshed. The question arises: does a rotation of assignment $X$ exist that guarantees robustness $R = 1$? As seen in Figure 3b, there is a rotation that avoids the loss of competence, and consequently holds robustness $R = 1$. This means that the appropriate rotation of teacher assignments affects the CSR in terms of particular disruptions (e.g., employee absenteeism).

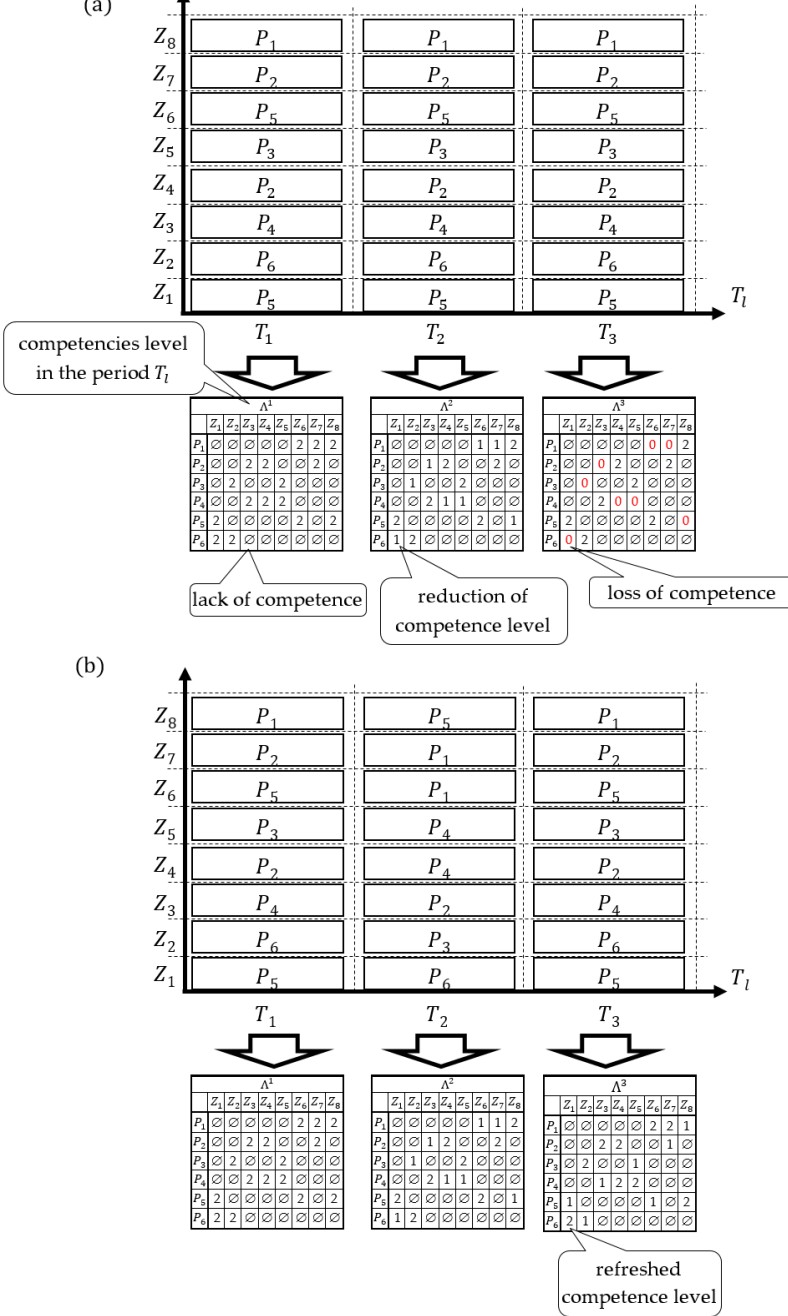

**Figure 3.** The assignment leading to loss of competencies (**a**), and the assignment with rotation which provides curriculum without losing teachers competencies (**b**).

The space of potential solutions grows exponentially when checking whether a given competence structure guarantees the expected level of robustness, and simultaneously

guaranteeing that the team's competence is kept unchanged. Greater robustness associated with greater mutual replacement of team employees reduces the chance of maintaining their competencies, i.e., the chances of finding appropriate staffing rotations (the number of employees with the same competencies increases, while the number of potential staffing does not change).

In general, solutions resulting in maximum robustness and guaranteeing the preservation of the team's competencies form multi-variant sets of competence structures. This means that the problem considered can be formulated either as a forward or reverse kind of problem.

In the first case, the problem is essentially checking if a given competence structure is able to guarantee the given robustness $R$ while maintaining the existing competencies of team members.

In the second, the answer is sought to these questions: Is there a competence structure that guarantees the given robustness and preserves the team's competencies? What structure guarantees the greatest robustness whilst maintaining the team's level of competence at a constant level?

The presented example takes into account the basic elements of the reference model of a proactive planning of CSR [10], such as:

- Characteristics of employees, performed activities (courses), assumed disruptions, etc.,
- Decision variables including competence robustness and a measure of its robustness to selected disruptions,
- A set of constraints (relationships connecting decision variables) characterizing the requirements in the field of competence structure and courses assignment.

This means that taking into account the forgetting effect requires the introduction of additional parameters, such as:

- Shapes of forgetting curves which specify the employees,
- Restrictions taking into account arbitrarily adopted competencies levels, the exceeding of which leads to a change in the $G$ competence of team members.

### 3.2. Problem Formulation

The case illustrated in the previous section can be summarized in the following problem: A university curriculum $Q$ is given. The courses in the curriculum are known and are represented by set $Z = \{Z_1, \ldots, Z_i, \ldots, Z_n\}$, where $Z_i$ is the $i$-th course of curriculum $Q$. Courses are executed cyclically in periods: $T_1, \ldots, T_l, \ldots, T_L$, where $L$ means the rotation cycle. It is assumed that in each period $T_l$ all courses from the set $Z$ are realised. Set $\mathcal{P} = \{P_1, \ldots, P_k, \ldots, P_m\}$ defines a team of teachers who have a set of competencies to conduct courses. The team of teachers $\mathcal{P}$ corresponds to a competence structure defined as a matrix $G$:

$$G = \left[g_{k,i}\right]_{k=1\ldots m; i=1\ldots n}, \tag{4}$$

where $g_{k,i} \in \{0,1\}$,

$$g_{k,i} = \begin{cases} 1 & \text{a teacher } P_k \text{ has the competencies to perform course } Z_i \\ 0 & \text{otherwise} \end{cases}$$

The competence's structure changes over time. The parameter $cl_{k,i}$ determines the competence lifetime after which the teacher $P_k$ loses (forgets) the competence $g_{k,i}$.

If teacher $P_k$ has the competence required to perform the course $Z_i$ ($g_{k,i} = 1$), then this course may be assigned to $P_k$. As a consequence, the assignment $X^l$ is created, which specifies which courses $Z$ are assigned to each teacher during the period $T_l$. This assignment is defined as the matrix $X^l$, whose elements $x_{k,i}$ are assigned with values $\{0,1\}$:

$$X^l = \left[x_{k,i}^l\right]_{k=1,\ldots,m; \, i=1,\ldots,n}, \tag{5}$$

where $x_{k,i}^l \in \{0,1\}$ specifies whether course $Z_i$ is performed during the period $T_l$ by $P_k$.

Moreover, each teacher $P_k$ is assigned a pair $\Gamma_k = (\overleftarrow{\gamma_k}, \overrightarrow{\gamma_k})$ which specifies the minimum ($\overleftarrow{\gamma_k}$) and maximum ($\overrightarrow{\gamma_k}$) number of courses assigned to them in period $T_l$.

The disruption (single employee absence), which is characterized by a sequence: $A = (a_1, .., a_l, ..., a_L)$, where $a_l \in \mathcal{P}$ determines the absent teacher in the period $T_l$. For example, $A = (P_1, P_2, P_1)$ means that teacher $P_1$ is absent during the first and third period $(T_1, T_3)$ and teacher $P_2$ is absent during the second period $(T_2)$.

According to (1) the measure of robustness of the competence structure $G$ to the absence of teachers is defined by function $R(A) = R \in [0,1]$, where:

- $R = 0$—lack of robustness, i.e., for each case of absenteeism there does not exist an assignment $X$ that guarantees the execution of curriculum $Q$;
- $R = 1$—full robustness, i.e., for each case of absenteeism there exists an assignment $X$ that guarantees the execution of curriculum $Q$.

The answer to the following questions is sought:

1. Can a given curriculum $Q$ be completed without losing competencies in the $G$ structure?
2. Does an assignment $X$ exist that guarantees a given value of robustness (e.g., $R = 1$)?
3. What is the maximum robustness $R$ of the competence structure $G$?

The introduction of additional assumptions allows for formulating further questions which may relate to competence structure, for example:

1. Which minimum competence structure (i.e., containing the minimum number of ones) guarantees maintaining a constant level of the team's competencies and robustness $R = 1$?
2. Which competence structure will give the maximum team robustness to teachers' absenteeism?

Other questions may relate to different variants of the assignment $X$ (e.g., are there assignments in subsequent periods that guarantee that the acquired competencies will not be lost for teachers $P_1$, $P_3$, $P_4$?) and to the competence lifetime (e.g., what is the optimal lifetime of the teacher's competencies (lowest value), which guarantees a constant level of competencies for the team?).

In the following section, a model will be presented which helps to determine the assignment $X = (X^1, ..., X^l, ..., X^L)$ and guarantees the execution of the curriculum $Q$ without losing teacher's competencies with maximum robustness $R$ of competence structure $G$.

## 4. Proactive Modelling Approach

Generally, in addition to the possibility of the organization characterized by the potential of its human resources (in particular its competence structure), the problem under consideration should take into account the expectations of the curriculum, in particular those related to the courses performed. This means that the considered problem could be described using the declarative modeling paradigm.

### 4.1. Reference Model

Sets:

$Z$:    a set of courses required to complete in curriculum $Q$: $Z = \{Z_1, ..., Z_i, ..., Z_n\}$,

$\mathcal{P}$:    a set of teachers, $\mathcal{P} = \{P_1, ..., P_k, ..., P_m\}$,

Parameters:

$n$:    number of courses executed in curriculum $Q$,

$m$:    number of teachers $\mathcal{P}$,

$L$:    rotation cycle,

$\Gamma_k$: pair $\Gamma_k = (\overleftarrow{\gamma_k}, \overrightarrow{\gamma_k})$ specifies the limits of the courses assigned to the teacher $P_k$,

$\overleftarrow{\gamma_k}$: minimum number of courses assigned to the teacher $P_k$ in one period $T_l$,

$\overrightarrow{\gamma_k}$: maximum number of courses assigned to the teacher $P_k$ in one period $T_l$,

$g_{k,i}$: competence of the teacher $P_k$ to perform course $Z_i$: $g_{k,i} \in \{0,1\}$,

$cl_{k,i}$: $g_{k,i}$ lifetime, $cl_{k,i} \in N$

$MC$: maximum competence lifetime: $MC = \max\limits_{k=1,\ldots,m; i=1,\ldots,n} \{cl_{k,i}\}$.

Decision variables:

$X^l$: assignment of courses $Z$ of curriculum $Q$ to the teachers $\mathcal{P}$ during the period $T_l$, $X^l = [x_{k,i}^l]_{k=1,\ldots,m;\ i=1,\ldots,n}$, where $x_{k,i}^l \in \{0,1\}$,

$Y^{l,\mu}$: assignment of courses $Z$ of curriculum $Q$ to the teachers $\mathcal{P}$ during the period $T_l$ in the case when a given teacher $P_\mu$ is absent, $Y^{l,\mu} = [y_{k,i}^{l,\mu}]_{k=1,\ldots,m;\ i=1,\ldots,n}$, where $y_{k,i}^{l,\mu} \in \{0,1\}$,

$W$: robustness matrix, $W = [w_{l,\mu}]_{l=1,\ldots,L;\ \mu=1,\ldots,m+L}$, where $w_{l,\mu} = 0$ when the competence structure $G$ is not robust to the absence of a given $P_\mu$, during the period $T_l$, in the other cases $w_{l,\mu} = 1$.

Constraints:

1. The teacher cannot perform a course for which they are not competent:

$$x_{k,i}^l \leq g_{k,i};\ k = 1,\ldots,m; i = 1,\ldots,n;\ l = 1,\ldots,(L+MC) \tag{6}$$

2. In each period $T_l$ all courses must be completed:

$$\sum_{k=1}^{m} x_{k,i}^l = 1;\ i = 1,\ldots,n;\ l = 1,\ldots,(L+MC) \tag{7}$$

3. In period $T_l$ numbers of courses assigned to $P_k$ are limited by $\Gamma_k = (\overleftarrow{\gamma_k}, \overrightarrow{\gamma_k})$:

$$\sum_{i=1}^{n} x_{k,i}^l \geq \overleftarrow{\gamma_k}; k = 1,\ldots,m;\ l = 1,\ldots,(L+MC) \tag{8}$$

$$\sum_{i=1}^{n} x_{k,i}^l \leq \overrightarrow{\gamma_k}; k = 1,\ldots,m;\ l = 1,\ldots,(L+MC) \tag{9}$$

4. The assignment $X = (X^1,\ldots,X^l,\ldots,X^L)$ for the curriculum execution should be cyclic (with cycle $L$):

$$x_{k,i}^l = x_{k,i}^{L+l};\ k = 1,\ldots,m; i = 1,\ldots,n;\ l = 1,\ldots,MC \tag{10}$$

5. Competencies should be refreshed in the competence lifetime $cl_{k,i}$:

$$\sum_{u=0}^{cl_{k,i}-1} x_{k,i}^{l+u} \geq g_{k,i};\ k = 1,\ldots,m; i = 1,\ldots,n;\ l = 1,\ldots,L \tag{11}$$

6. Courses are not assigned to absent teacher $P_\mu$:

$$y_{k,i}^{l,\mu} = 0;\ k = \mu;\ k,\mu = 1,\ldots,m; i = 1,\ldots,n;\ l = 1,\ldots,(L+MC) \tag{12}$$

7. When teacher $P_\mu$ is absent, other teachers provide the assigned courses:

$$y_{k,i}^{l,\mu} \geq x_{k,i}^l;\ k \neq \mu;\ k,\mu = 1,\ldots,m; i = 1,\ldots,n;\ l = 1,\ldots,(L+MC) \tag{13}$$

8. In the case when teacher $P_\mu$ is absent in each period $T_l$, all courses should be executed:

$$\sum_{k=1}^{m} y_{k,i}^{l,\mu} = 1;\ \mu = 1,\ldots,m\ \ i = 1,\ldots,n;\ l = 1,\ldots,(L+MC) \tag{14}$$

9. In the case when teacher $P_\mu$ is absent, the assignment $Y^\mu = (Y^{1,\mu},\ldots,Y^{l,\mu},\ldots,Y^{L,\mu})$ for the curriculum execution should be cyclic (with cycle $L$):

$$y_{k,i}^{l,\mu} = y_{k,i}^{L+l,\mu};\ k,\mu = 1,\ldots,m; i = 1,\ldots,n;\ l = 1,\ldots,MC \tag{15}$$

10. If the replacement $P_\mu$ requires additional competencies (for other teachers), then the competence structure is not robust to the absence of this teacher ($w_{l,\mu} = 0$):

$$\left(y_{k,i}^{l,\mu} > g_{k,i}\right) \Rightarrow (w_{l,\mu} = 0); \quad k, \mu = 1, \dots, m; i = 1, \dots, n; \ l = 1, \dots, (L + MC) \quad (16)$$

11. If the replacement $P_\mu$ requires exceeding the limits of courses $\overrightarrow{\gamma_k}$ assigned to $P_k$, then the competence structure is not robust to the absence of this teacher ($w_{l,\mu} = 0$):

$$\left(\sum_{i=1}^{n} y_{k,i}^{l,\mu} \geq \overleftarrow{\gamma_k}\right) \Rightarrow (w_{l,\mu} = 0); \quad k, \mu = 1, \dots, m; \ l = 1, \dots, (L + MC) \quad (17)$$

12. The number of disruption scenarios to confirm the competence structure is robust is calculated as a sum of values of $w_{l,\mu}$:

$$LP = \sum_{l=1}^{L+MC} \sum_{\mu=1}^{m} w_{l,\mu} \quad (18)$$

Objective function:

Maximize the robustness $R$ of competence structure $G$:

$$maximize: R = \frac{LP}{m \times (L + MC)} \quad (19)$$

The presented model allows to answer the following question: What is the maximum robustness $R$ of the competence structure $G$ without losing competencies?

The above problem can be formulated as a COP (constraint optimization problem) and takes the following form:

$$CO = \left((\mathcal{V}, \mathcal{D}), \mathcal{C}, \mathcal{C}_{OPT}\right) \quad (20)$$

where
$\mathcal{V} = \left\{x_{k,i}^l, y_{k,i}^{l,\mu} | k, \mu = 1, \dots, m; i = 1, \dots, n; \ l = 1, \dots, (L + MC)\right\}$, a set of decision variables representing assignment: $X = (X^1, \dots, X^l, \dots, X^L)$ and $Y^\mu = (Y^{1,\mu}, \dots, Y^{l,\mu}, \dots, Y^{L,\mu})$;
$\mathcal{D}$ is a finite set of domains of decision variables: $x_{k,i}^l \in \{0,1\}$, $y_{k,i}^{l,\mu} \in \{0,1\}$
$\mathcal{C}$ is a set of constraints specified in inequalities (6)–(18).
$\mathcal{C}_{OPT}$ is a constraint specifying the objective function (19).

To solve problem $CO$ (20), one should determine such values of decision variables $x_{k,i}^l$, $y_{k,i}^{l,\mu}$ (assignment), for which all the constraints given in the set $\mathcal{C}$ are satisfied and the objective function (18) is the maximum. Solving $CO$ means determining the assignment which guarantees the curriculum $Q$ execution with maximum robustness and without losing employees' competencies. It is worth noting that the requested assignment determines the rotation cycle $L$, thus guaranteeing job rotation.

*4.2. Sufficient Conditions*

Due to the fact that the number of possible initial state schedules is determined by the permutation with repetitions: $P(m, n) = m^n$, (where $m$ is the number of teachers, $n$ is the number of courses), the considered problem belongs to the class of problems NP-complete. This means that it is necessary to specify conditions that significantly limit the space of possible solutions, in particular sufficient conditions, e.g., taking the form of the following inequalities:

$$\sum_{k=1}^{m} g_{k,i} \leq \min_{k=1\dots m}\{cl_{k,i}\}, \ i = 1, \dots, n \quad (21)$$

$$\sum_{i=1}^{n} g_{k,i} \leq \overrightarrow{\gamma_k} \times \min_{i=1\dots n}\{cl_{k,i}\}, \ k = 1, \dots, m \quad (22)$$

However, the introduction of such constraints does not guarantee the existence of an admissible (cyclical) solution. This means that a competence structure $G$ may exist for which no assignment $(X^1, \dots, X^l, \dots, X^L)$ can be found that guarantees the implementation

of the curriculum $Q$ without losing teachers' competencies. An extensive discussion was undertaken in [49], which shows that the proactive maintenance of employee competencies requires the assignment of rotation plans. The example of using our approach is presented in the next section.

## 5. Case Study

This case study describes the teachers assignment process at the Koszalin University of Technology (KUT) in the 2021–2022 academic year. The university carries out educational activities and scientific research in disciplines related primarily to the development of the Polish Middle Pomerania region.

KUT offers students 24 programs (full-time and part-time) at both graduate and undergraduate levels. In general, student education takes place at six faculties. The data used to carry out the conducted experiments was obtained from the Faculty of Electronics and Computer Science (FECS).

The process of organizing the courses adopted at FECS consisted of three stages:

1. Defining the requirements: The FECS curriculum in the 2021–2022 academic year included $n$ = 129 courses: $\mathcal{Z} = \{Z_1, Z_2, \ldots, Z_{129}\}$ (for BSc and MSc courses), with a total of 3800 h. The components of the courses $Z_i$ are shown in Table 3.

**Table 3.** Faculty of Electronics and Computer Science (FECS) curriculum.

| Courses $Z_i$ |
|---|
| $Z_1$: History of technics 1 |
| $Z_2$: History of technics 2 |
| $Z_3$: Inventics |
| $Z_4$: Economics |
| … |
| $Z_{74}$: Programming in. NET environment |
| … |
| $Z_{128}$: Distributed information processing systems |
| $Z_{129}$: Artificial intelligence methods |

2. Assessment of capabilities: In the 2021–2022 academic year, 32 teachers $\mathcal{P} = \{P_1, P_2, \ldots, P_{32}\}$ were employed at FECS. For each of them, their competencies (education, scientific achievements, knowledge of a given course, etc.) were known, which defined the courses that they could conduct. Table 4 presents the components of the competence structure $G$. The value of 1 means that the teacher had the competence to teach a specific course, the value of 0 represents the opposite.

**Table 4.** Competence structure $G$ of FECS teaching staff.

| $G$ | $Z_1$ | $Z_2$ | $Z_3$ | $Z_4$ | $Z_5$ | $Z_6$ | … | $Z_{60}$ | … | $Z_{128}$ | $Z_{129}$ |
|---|---|---|---|---|---|---|---|---|---|---|---|
| $P_1$ | 1 | 0 | 0 | 1 | 0 | 0 | … | 0 | … | 0 | 1 |
| $P_2$ | 0 | 1 | 0 | 0 | 1 | 0 | … | 0 | … | 0 | 0 |
| $P_3$ | 1 | 0 | 1 | 0 | 0 | 0 | … | 0 | … | 0 | 0 |
| $P_4$ | 0 | 0 | 0 | 1 | 0 | 0 | … | 1 | … | 0 | 0 |
| … | … | … | … | … | … | … | … | … | … | … | … |
| $P_{20}$ | 0 | 0 | 0 | 0 | 0 | 0 | … | 1 | … | 0 | 0 |
| … | … | … | … | … | … | … | … | … | … | … | … |
| $P_{31}$ | 0 | 0 | 0 | 0 | 0 | 0 | … | 0 | … | 0 | 0 |
| $P_{32}$ | 0 | 0 | 0 | 0 | 0 | 0 | … | 0 | … | 0 | 0 |

3. Teachers' assignment: During this stage, teachers were assigned to the courses under the following given requirements:

- each course could be executed by only one competent teacher,
- all courses were executed in the same time period (academic year) $T_i$,
- each teacher had to perform a minimum of one course ($\overleftarrow{\gamma_k} = 1$), and a maximum of ten ($\overrightarrow{\gamma_k} = 10$) courses in the period $T_l$.

According to the forgetting effect, the competence lifetime for each teacher was equal to: $cl_{k,i} = 4$.

The answers for the following questions were sought:

1. Is there an assignment $X$ that guarantees robustness $R = 1$ to any single teacher absence without a loss of competencies?
2. What is the maximum robustness $R$ to any single teacher absence without losing competencies?

The answer (solving the $CO$ (20) problem) was obtained in the IBM ILOG CPLEX environment (Intel i7-10510U, 16 GB RAM) in 696 s.

A schedule of cyclic ($L = 4$) rotation of a teachers' assignment to courses was obtained (part of the schedule presented in Figure A1 in the Appendix A). It guaranteed the maintenance of all competencies in the competence structure $G$. For example, for employee $P_{16}$ to maintain competency related to the course $Z_{18}$, they were assigned to this course in periods $T_1$ and $T_5$, for employee $P_{28}$ to maintain competency related to the course $Z_{15}$, they were assigned to this course in periods $T_4$ and $T_8$, etc.

The robustness $R = 1$ was determined. This means that in each scenario of the absence of a single teacher ($U = 128$) substitutions were possible, for example:

- a teacher $P_1$ who executes a $Z_4$ course during the $T_1$ may be replaced by $P_4$,
- a teacher $P_2$ who executes a $Z_5$ course during the $T_1$ may be replaced by $P_5$,
- a teacher $P_{31}$ who executes a $Z_{12}$ course during the $T_2$ may be replaced by $P_{27}$,
- etc.

The answers to the questions raised earlier were as follows:

1. There was an assignment $X$ that guaranteed robustness $R = 1$.
2. Maximum robustness to single teacher absenteeism was $R = 1$.

For the decision-makers of the Faculty of Electronics and Computer Science, this meant that the competencies of the teachers employed were secured for 100% of the types of disruption considered.

## 6. Computational Experiments

The proposed approach was verified in two quantitative experiments. They adopted different scales of the problem (number of teachers and courses). In addition, the competence structures were randomly generated in the MATLAB environment, assuming:

- A given $G$ structure's compactness $CM$ determines the degree of competence saturation of a team:

$$CM = \frac{\sum_{k=1}^{m} \sum_{i=1}^{n} g_{k,i}}{n \times m} \tag{23}$$

For example, structure $G$ with ten teachers and fifty tasks had a total number of 500 competencies $g_{k,i}$ and compactness $CM = 40\%$, it meant that 200 competencies $g_{k,i}$ should have the value of 1.

- There was no teacher in the competence structure without a single competence and no course without a competent teacher.

### 6.1. Experiment 1

First, the scalability of different instances of the problem was considered, assuming: The set of teachers: $\mathcal{P} = \{P_1, \ldots, P_7\}$, $\mathcal{P} = \{P_1, \ldots, P_8\}$, $\mathcal{P} = \{P_1, \ldots, P_{10}\}$,

- The set of courses in the curriculum $Q$: $Z = \{Z_1, \ldots, Z_{10}\}$, $Z = \{Z_1, \ldots, Z_{12}\}$, $Z = \{Z_1, \ldots, Z_{14}\}$, …, $Z = \{Z_1, \ldots, Z_{28}\}$, $Z = \{Z_1, \ldots, Z_{30}\}$,
- Compactness $CM$ = 40%.

Answers to the following questions were sought:

1. What is the maximum robustness $R$ of competence structure $G$ without loss of competencies?
2. What time is needed for calculations?
3. What is the minimum rotation cycle $L$?

The calculations were carried out in the IBM ILOG CPLEX environment. The results are given in Table 5 and Figure 4. Maximum robustness was obtained for all variants of the problem scale $R = 1$. With the increasing scale the computation time increased. For example, when comparing the cases in which there are 30 courses, an increase in the number of $m$ teachers from seven to ten caused a twofold extension of the calculation time. In turn, increasing the number of $m$ teachers by two (from seven to nine) caused an extension of the rotation cycle $L$ by an average single period.

**Table 5.** Calculation time for the different scales of the competence structure with compactness $CM$ = 40%, and for competence lifetime $cl_{k,i}$ = 8.

| Number of Teachers $m$ | Number of Courses $n$ | Robustness $R$ | Minimum Rotation Cycle $L$ | Calculation Time [s] |
|---|---|---|---|---|
| 7 | 10 | 1 | 5 | 2.2 |
| 7 | 12 | 1 | 4 | 2.15 |
| 7 | 14 | 1 | 4 | 2.29 |
| 7 | 16 | 1 | 4 | 2.48 |
| 7 | 18 | 1 | 4 | 2.52 |
| 7 | 20 | 1 | 4 | 2.81 |
| 7 | 22 | 1 | 4 | 2.77 |
| 7 | 24 | 1 | 4 | 2.91 |
| 7 | 26 | 1 | 4 | 3.15 |
| 7 | 28 | 1 | 4 | 3.36 |
| 7 | 30 | 1 | 4 | 3.37 |
| 9 | 10 | 1 | 6 | 2.93 |
| 9 | 12 | 1 | 6 | 3.08 |
| 9 | 14 | 1 | 6 | 3.28 |
| 9 | 16 | 1 | 5 | 3.5 |
| 9 | 18 | 1 | 5 | 3.69 |
| 9 | 20 | 1 | 5 | 3.96 |
| 9 | 22 | 1 | 5 | 4.28 |
| 9 | 24 | 1 | 5 | 4.64 |
| 9 | 26 | 1 | 5 | 4.88 |
| 9 | 28 | 1 | 5 | 4.91 |
| 9 | 30 | 1 | 5 | 5.25 |
| 10 | 10 | 1 | 6 | 3.81 |
| 10 | 12 | 1 | 5 | 3.94 |
| 10 | 14 | 1 | 6 | 4.12 |
| 10 | 16 | 1 | 5 | 4.04 |
| 10 | 18 | 1 | 5 | 4.34 |
| 10 | 20 | 1 | 5 | 5.01 |

| 10 | 22 | 1 | 5 | 5.05 |
| 10 | 24 | 1 | 5 | 5.41 |
| 10 | 26 | 1 | 5 | 5.69 |
| 10 | 28 | 1 | 5 | 6.13 |
| 10 | 30 | 1 | 5 | 6.7 |

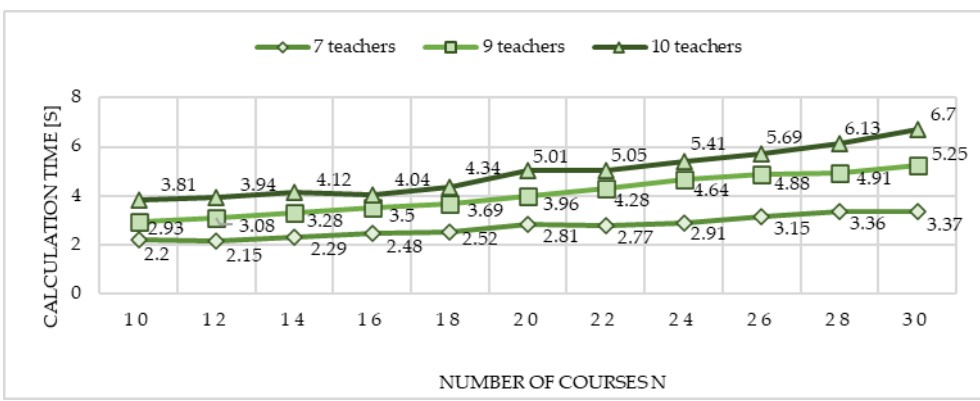

**Figure 4.** The comparison of the calculation time for various number of teachers $m$.

*6.2. Experiment 2*

In the second experiment, the impact (on the calculation time and minimum rotation cycle $L$) of different compactness of the competence's structure ($CM$ = 40%, $CM$ = 50%, $CM$ = 60 %) was examined. It was assumed that the number of courses was twice the number of teachers. The results are shown in Table 6, and the comparison of calculation times is illustrated in Figure 5.

**Table 6.** Various compactness $CM$ impact analysis (competence lifetime $cl_{k,i}$ = 6).

| Number of Teachers $m$ | Number of Courses $n$ | Compactness $CM$ | Minimum Rotation Cycle $L$ | Calculation Time [s] |
|---|---|---|---|---|
| 5 | 10 | 40% | 3 | 3.26 |
| | | 50% | 3 | 3.29 |
| | | 60% | 4 | 4.99 |
| 6 | 12 | 40% | 3 | 3.98 |
| | | 50% | 4 | 5.83 |
| | | 60% | 5 | 7.74 |
| 7 | 14 | 40% | 3 | 4.87 |
| | | 50% | 5 | 9.54 |
| | | 60% | 5 | 9.86 |
| 8 | 16 | 40% | 4 | 9.32 |
| | | 50% | 5 | 12.03 |
| | | 60% | 6 | 17.61 |
| 9 | 18 | 40% | 5 | 14.69 |
| | | 50% | 6 | 19.61 |
| | | 60% | 6 | 20.82 |
| 10 | 20 | 40% | 5 | 19.31 |
| | | 50% | 6 | 26.72 |
| | | 60% | 6 | 28.16 |

It can be observed that with an increase in the $CM$, the following impacts appeared:

- The computation time increased, for example, for eight courses and 16 teachers, the difference in the calculation time between $CM$ = 40% and $CM$ = 60% was about 8 s,

- The rotation cycle $L$ increased only for some cases, for example, for eight courses and 16 teachers, $L = 4$ for $CM = 40\%$, $L = 5$ for $CM = 50\%$, $L = 6$ for $CM = 60\%$.

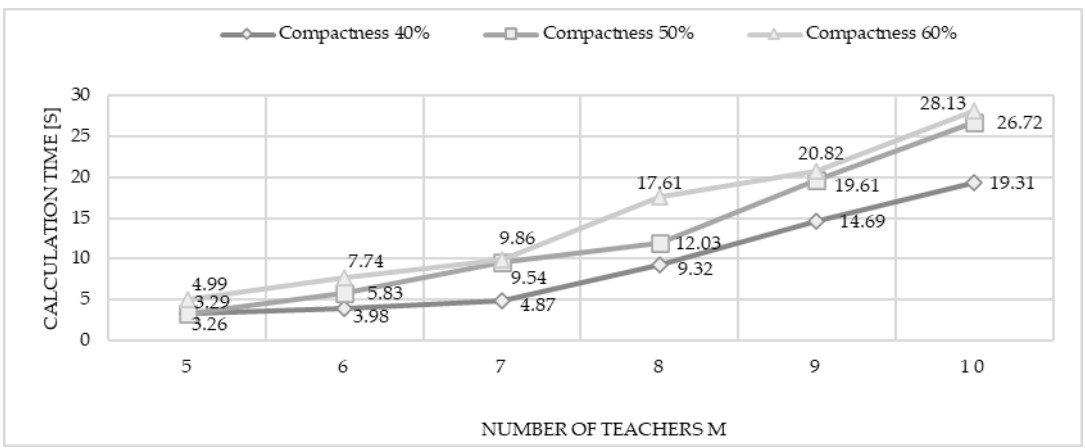

**Figure 5.** The comparison of the calculation times for various compactness $CM$.

The most important observation in Table 6 is that if the minimum rotation cycle $L$ extends with the increase of compactness of the competency's structure $CM$, then the differences in computation times are significant, e.g., in case of eight teachers and 16 courses, between $CM = 40\%$ ($L = 4$) and $CM = 50\%$ ($L = 5$), the difference was 2.71 s, between $CM = 50\%$ ($L = 5$) and $CM = 60\%$ ($L = 6$), the difference was 5.58 s. Otherwise, such as in case of nine teachers and 18 courses, between $CM = 40\%$ ($L = 5$) and $CM = 50\%$ ($L = 6$), the difference was 4.92 s, and between $CM = 50\%$ ($L = 6$) and $CM = 60\%$ ($L = 6$), the difference was only 1.21 s. This experiment shows that the time to solve a problem is not impacted so much by $CM$ as by the minimum rotation cycle $L$.

## 7. Conclusions

In this paper, the emphasis was placed on the problem of planning the rotational assignment of work tasks to a multi-skilled staff to guarantee maintaining their competencies at the required level. A novel declarative model for proactive planning of staff allocation whilst taking into account the forgetting effect was proposed. The main conclusions from the presented research are as follows:

(1) The reference model including the forgetting effect resulting in the loss of previously acquired competencies allows the formulation of the problem of maintaining human resources (MHR) in a manner similar to the problem of maintaining the movement of machines in production processes. The solution to the problem of MHR perceived in this way is a plan of periodic rotation of the workstations and activities that allows employee competencies to be maintained at the required level of robustness. It is easy to see that the proposed extension of the reference model forces a significant increase in the calculation expenditure incurred in the process of planning the staff allocation to tasks or courses.

(2) The implementation of the proposed approach implies the emergence of a new class of trade-off problems, in which the desired rotation of staff is conditioned on the one hand by the time of task execution, and on the other hand by the validity of competencies. Human resource management systems take into account artificial intelligence trends such as data mining and machine learning, cloud-based solutions, agent-based skills and knowledge management systems. So far, however, there is no information about solutions that take into account the needs related to maintaining the competencies of multi-skilled teams in situations related to the forgetting effect.

(3) The experiments and case study have shown that the proposed approach can be used online in practice, i.e., with the curriculum of 120 courses and teams of 30 teachers.

Therefore, this paper has managerial implications. The results of this study can be used in decision support systems to maintain employees' competencies through appropriate job rotation and therefore more sustainable human resource planning (development of long-term competencies, employee empowerment, reduced fatigue, and boredom, etc.).

The main limitation of this study results from the fact that only one learning and forgetting model (LFCM) was considered. Comparing the proposed approach with other models such as RC and PID should be the subject of future research. Future studies should take into account the different shapes of the forgetting function characteristic of different professions and age groups.

**Author Contributions:** Conceptualization, E.S., G.B., and Z.B.; methodology, ES. and G.B.; software, G.B.; validation, E.S.; formal analysis, G.B.; investigation, E.S.; resources, Z.B.; data curation, E.S. and P.G.-D.; writing—original draft preparation, P.G.-D. and E.S.; writing—review and editing, Z.B. and P.G.-D.; visualization, G.B. and E.S.; supervision, Z.B.; project administration, Z.B. and P.G.-D. All authors have read and agreed to the published version of the manuscript.

**Funding:** This research received no external funding.

**Institutional Review Board Statement:** Not applicable.

**Informed Consent Statement:** Not applicable.

**Data Availability Statement:** Not applicable.

**Conflicts of Interest:** The authors declare no conflict of interest.

**Appendix A**

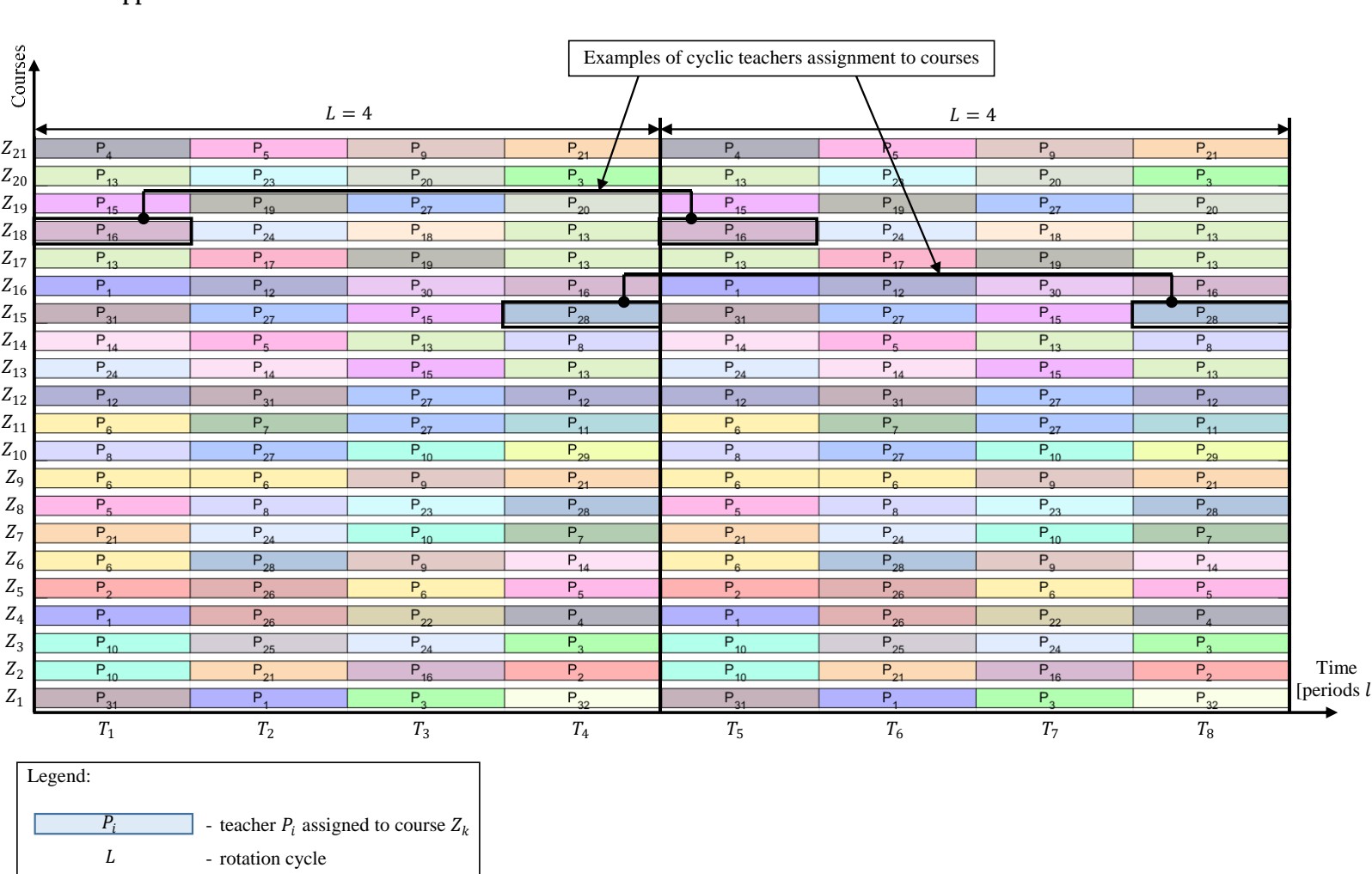

**Figure A1.** Part of the schedule of cyclic ($L = 4$) rotation of teacher assignment $X$ guaranteeing the maintenance of all competencies.

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
