# Peer review of "Proactive Operations Management: Staff Allocation with Competence Maintenance Constraints"

_sustainability, doi:10.3390/su15031949_

Round 1

Reviewer 1 Report

Recommendations for the authors of the article:

1. Add a section: "Literature review". Particular attention should be paid to the conditions for the development of human capital management competencies.

2. The article does not pay enough attention to the role of intellectual assets in building the competitiveness of modern economies.

3. To improve the scientific accuracy of the article and to refer to the proposed changes, I propose to include the following items: https://doi.org/10.3390/risks9060107, https://doi. org/10.1016/j.tree.2012.09.003, https://doi.org/10.1108/09696470410515715

4. In the article the conclusions of the studies should be given in sub-paragraphs. In addition, there is no description of the research limitations.

5. Finally, it is necessary to emphasize the postulated and accepted directions of action for strengthening social capital in the studied population.

Author Response

The authors thank the reviewers for very helpful comments and suggestions. The authors have incorporated the reviewers’ comments in the revised manuscript.

Responses to the Reviewers’ comments

Reviewer #1:

Remark 1: Add a section: "Literature review". Particular attention should be paid to the conditions for the development of human capital management competencies.

Response: The section ‘Related works’ was renamed ‘Literature review’, as suggested by the reviewer. This section was written based on keywords with focus on maintaining the competence of the staff, and staff scheduling. In Section 2.3. the research gap was highlighted. The proposed model contributes to the identified research gap. The suggested by the reviewer research on "conditions for the development of human capital management competencies" are interesting, but they are out of scope of this paper. In this paper, the emphasis is put on the problem of planning the cyclical assignment of courses to a multi-skilled staff to guarantee the implementation of courses without losing the competencies already possessed by a given employee.

To highlight the human capital management issues in maintaining competencies, the following excerpt was introduced into Section 2:

Page 3: “The changing conditions of the organization's operation, in particular those related to the pandemic period, underlined the value of managers' competencies related to the assessment of the risk of human resources availability”.

Remark 2: The article does not pay enough attention to the role of intellectual assets in building the competitiveness of modern economies.

Response: In the context of our response to Remark 1, we would like to note that it was also not our goal to focus on the role of intellectual assets and, in particular, issues related to building the competitiveness of modern economies. In our work, we focused on the technical skills and competencies obtained by employees and their maintenance opportunities throughout their lifetime.

The publication proposed for citation in Remark 3 (The Empirical Analysis of the Core Competencies of the Company's Resource Management Risk. Preliminary Study) in its title referring to the core competencies of the company's resource management, also fails to mention intellectual assets, and the intellectual property that comprises them.   

Remark 3: To improve the scientific accuracy of the article and to refer to the proposed changes, I propose to include the following items: https://doi.org/10.3390/risks9060107, https://doi. org/10.1016/j.tree.2012.09.003, https://doi.org/10.1108/09696470410515715

Response: Thank you for the suggestions. Works:

  • https://doi. org/10.1016/j.tree.2012.09.003,
  • https://doi.org/10.1108/09696470410515715,

were not included in the revised version of the manuscript because, according to the comment Reviewer #2, they belong to the category of old references.

Paper: https://doi.org/10.3390/risks9060107 was quoted in Section 2, with reference at position 15.

  1. Drozdowski, G.; Rogozinska-Mitrut J.; Stasiak J. The Empirical Analysis of the Core Competencies of the Company’s Resource Management Risk. Preliminary Study. Risks 2021; 9: 107.

Remark 4: In the article the conclusions of the studies should be given in sub-paragraphs. In addition, there is no description of the research limitations.

Response: As suggested, Section 7 was divided into subsections. Issues related to the description of the research limitations were addressed in Section 3.1, and in Section 3.2 (when formulating the assumptions of the model and the problem based on it). Furthermore, they were described also in the conclusion of Section 7 (when stating directions for the future research).

Remark 5: Finally, it is necessary to emphasize the postulated and accepted directions of action for strengthening social capital in the studied population.

 Response: With regard to the responses to Remarks 1 and 2, we would like to emphasize that it was not our goal to postulate and propose directions of action for strengthening social capital in the population under study.

We would like to note that our article is neither a review paper nor a paper reporting the results of surveys (of the Empirical Analysis type, the results of which are limited to selected populations, environments, etc.). We propose an analytical model that allows for quantitative evaluation of the decisions made, as well as conducting calculations (related to determining the values of decision variables) that determine their success (synthesis class problems).

Reviewer 2 Report

Thanks for the opportunity to review the manuscript " Proactive operations management - staff allocation with competence maintenance constraint’’ by Eryk Szwarc et al. The work is interesting. Possibly, it could be useful to the wider research community. I have gone through the complete manuscript.

1.         Novelties are not convincing. Author must serious about the novelty in the manuscript.

2.         Keywords are too large.

3.         The Abstract, and Conclusion should be more informative and concise.

4.         There are several errors, grammar, construction, etc. of the manuscript.

5.         Don’t reflect the entire finding of the research. Please update significantly.

6.         Structure of the paper can be more attractive.

7.         Try to avoid large citation. Delete the old references.

You may refer this paper.

A multi-item multi-objective inventory model in exponential fuzzy environment using chance-operator techniques

Best wishes

Author Response

The authors thank the reviewers for very helpful comments and suggestions. The authors have incorporated the reviewers’ comments in the revised manuscript.

Responses to the Reviewers’ comments

Remark 1: Novelties are not convincing. Author must serious about the novelty in the manuscript.

Response:

We have highlighted the novelty of this paper by adding in Introduction on page 2, the text as follows:

In this paper, the emphasis is put on the problem of planning the cyclical assignment of courses to a multi-skilled staff to guarantee the implementation of courses without losing the competencies already possessed by a given employee. This paper aims to propose a novel declarative model for proactive planning of staff rotational allocation whilst taking into account the effect of forgetting.

Previous studies do not propose comprehensive models for balancing available multi-skilled employees with the requirements resulting from workspan. Furthermore, there is a lack of proactive planning methods for the assignment and rotation of a team of employees (through synthesis of the suitable competence structure) when ad hoc disruptions occur. There is a need for analytical models in the area of human resources which enable efficient rotation for maintaining the staff competences at a required level. A comprehensive literature and detailed definition of the research gap are provided in Section 2.”

In Section 2, we identified the research gap:

“The following essential research gaps can be summarized from our literature re-view:

  • There are no comprehensive models that would enable the study to balance available human resources (in particular, multi-skilled staff members) with the requirements resulting from particular orders,
  • There is lack of methods of proactive management of human resources, in particular proactive planning of assignment and rotation of employee tasks, i.e., enabling the formation of employee teams (through synthesis of the suitable competence structure) guaranteeing the timely execution of orders in situations related to ad hoc occurring disruptions,
  • There is lack of analytical models for maintaining multi-skilled human resources that would allow for the development of job (task) rotation methods ensuring the maintenance of staff competences at a constant level.”.

In the Section “Introduction” we described the source of the novelty of this paper: 

The novelty of this study results from:

  • Proposing of generic model for proactive allocation of multi-skill workforce whilst taking into account forgetting effect,
  • Defining sufficient conditions for cyclical relocations of employee to maintain their competence at a constant level,
  • Developing a method for selecting the competencies of team members aimed at a trade-off between the assumed robustness to absenteeism and maintenance of the competences of team members by cyclical rotation of positions.”.

If the above justification is not sufficiently described, then please indicate which of the above-mentioned gaps and/or novelties require additional commentary.

Remark 2: Keywords are too large.

Response: Thank you for the comment. Two keywords were removed.

Remark 3: The Abstract, and Conclusion should be more informative and concise.

Response: Thank you for the comment. We have rewritten the Abstract, and the Conclusions to make them more informative and concise.

Remark 4: There are several errors, grammar, construction, etc. of the manuscript.

Response: Thank you for bringing this to our attention. In addition to the autocorrection of the text, the manuscript has undergone additional proofreading.

Remark 5: Don’t reflect the entire finding of the research. Please update significantly.

Response: As in Remark 3, the reviewer assessed that "arguments and discussion of findings are coherent, balanced and compelling," while in this descriptive remark he points out the lack of research results. We therefore kindly ask you to be more specific/specific in your comments.

We would also like to note that in Section 2, while pointing out the research gaps that have been observed in the literature, it was noted that:

“The following essential research gaps can be summarized from our literature re-view:

  • There are no comprehensive models that would enable the study to balance available human resources (in particular, multi-skilled staff members) with the requirements resulting from particular orders,
  • There is lack of methods of proactive management of human resources, in particular proactive planning of assignment and rotation of employee tasks, i.e., enabling the formation of employee teams (through synthesis of the suitable competence structure) guaranteeing the timely execution of orders in situations related to ad hoc occurring disruptions,
  • There is lack of analytical models for maintaining multi-skilled human resources that would allow for the development of job (task) rotation methods ensuring the maintenance of staff competences at a constant level.”.

Furthermore, the main contributions have been indicated in the Introduction section.: 

The novelty of this study results from:

  • Proposing of generic model for proactive allocation of multi-skill workforce whilst taking into account forgetting effect,
  • Defining sufficient conditions for cyclical relocations of employee to maintain their competence at a constant level,
  • Developing a method for selecting the competencies of team members aimed at a trade-off between the assumed robustness to absenteeism and maintenance of the competences of team members by cyclical rotation of positions.”.

If the above justification is not sufficiently described, then please indicate which of the above-mentioned gaps and/or novelties require additional commentary.

Remark 6: Structure of the paper can be more attractive.

Response: The structure of the article is typical of the research being conducted (e.g. http://dx.doi.org/10.17531/ein.2021.1.13.). However, to improve the its attractiveness, the section 4.2 has been  rewritten.

In case this proves insufficient, please indicate and suggest which sections in what way should be changed (please give a typical example of this).

Remark 7: Try to avoid large citation. Delete the old references.

Response: Out of a total of 51 items of literature, 17 of them are older than 2015, which is a small percentage of the bibliography. Some of them constitute the canon of literature, e.g., on research on the effect of forgetting (References 33-36). Nevertheless, 4 papers prior to 2015 were removed. If, in the reviewer's opinion, there are other redundant ones, please indicate them.

Reviewer 3 Report

The authors present an interesting topic. The paper is well organized, the literature is relevant and the proposed model seems sufficiently reach the stated research objectives.

However, there are some elements worth expanding given the scope of the work. They are discussed below.

The competence is discussed, but it is not clear how the strength of knowledge is taken into consideration. One cannot assume that the experts have a perfect knowledge in their field of expertise. The authors should clarify how the strength of knowledge is incorporated into the model.

Uncertainties related to variables used: it seems that uncertainties related to the variables in the model are not specifically taken into account. The approach is far too deterministic and linear. The authors should explain this weakness.

Some discussion upon the topic regarding it as a complex adaptive system is needed given that the analyzed problem should be seen as such. There are so many variables that interact between them and it is quite cumbersome to build an accurate analytic model. The methods and tools of the Complexity Science (or Complexity Theory) could greatly improve the modeling of such systems. For that reason, the authors should provide some insights/discussion and propose future research.

The limits of the proposed model should be presented in a more detailed way.

Author Response

The authors thank the reviewers for very helpful comments and suggestions. The authors have incorporated the reviewers’ comments in the revised manuscript.

 Responses to the Reviewers’ comments

Reviewer #3:

Remark 1: The competence is discussed, but it is not clear how the strength of knowledge is taken into consideration. One cannot assume that the experts have a perfect knowledge in their field of expertise. The authors should clarify how the strength of knowledge is incorporated into the model.

Response: The proposed model assumes "zero-one" states of the level of professional-methodical competencies, which a person needs in order to solve problems. The model can be extended in the future, for example, in terms of application of discrete fuzzy numbers for assessing in scale such, as the type 0 - ¼ - ½ - ¾ - 1. The adopted assessment of professional knowledge, skills and methods depends on the profession involved and the manner in which it is measured. In this work we have adopted arbitrary assessments of the manager. 

However, these issues have been not developed further in this work, but they were discussed for example in works:

  • Method to measure competencies - a concept for development, design and validation, Glass, Rupert, AU  - Metternich, Joachim, Procedia Manufacturing https://doi.org/10.1016/j.promfg.2020.04.056
  • On the measurement of competency July 2010 Empirical Research in Vocational Education and Training2(1):41-63 DOI: 1007/BF03546488
  • Rohr-Mentele, S., Forster-Heinzer, S. Practical validation framework for competence measurement in VET: a validation study of an instrument for measuring basic commercial knowledge and skills in Switzerland. Empirical Res Voc Ed Train13, 18 (2021). https://doi.org/10.1186/s40461-021-00122-2

Remark 2: Uncertainties related to variables used: it seems that uncertainties related to the variables in the model are not specifically taken into account. The approach is far too deterministic and linear. The authors should explain this weakness.

Response: In the proposed model, uncertainty of parameters is not taken into account. Uncertainty in the considered class of problems may concern the values of adopted variables (e.g., competence lifetime, working time limits, etc.), as well as disruptions related to employee fluctuations (e.g., absences, departures from work, etc.), variable number of tasks, changes in customer requirements, etc. The proposed approach assumes that the adopted parameter values are described by precise values representing a pessimistic scenario (e.g., the shortest lifetime of competencies). Thus, the results obtained are excessive - they guarantee a predetermined robustness in the worst-case scenario.

Due to the open structure of the proposed model, it is easy to extend it with fuzzy representations as well as non-linear constraints. They can take into account, among other things, the specifics of the job positions, the age of the employees, gender and their individual predispositions.

Extension of the developed model (construction of variants of the model dedicated to different applications), will be the subject of future research.  

Remark 3: Some discussion upon the topic regarding it as a complex adaptive system is needed given that the analyzed problem should be seen as such. There are so many variables that interact between them and it is quite cumbersome to build an accurate analytic model. The methods and tools of the Complexity Science (or Complexity Theory) could greatly improve the modeling of such systems. For that reason, the authors should provide some insights/discussion and propose future research.

Response: The proposed model provides a basis for various types of extension. The model is expressed (and implemented) in the terminology of declarative programming techniques (in particular, techniques with constraints). Its open structure allows modification as well as taking into account new relations occurring between decision variables; this is done without loss of computational efficiency (in constraints programming environments, i.e. IBM ILOG, increasing the number of constraints reduces the time of determining the solution). Additional constraints can relate to both the way competencies are modeled (single/multivalued/fuzzy representation), the characteristics of the work environment (the way tasks are performed, working in groups, the amount and type of resources used) and the nature of the disruptions occurring (employee absenteeism, loss of competencies, change in organizational structure, etc.).

Remark 4: The limits of the proposed model should be presented in a more detailed way.

Response: The model's assumptions are illustrated in Section 3.1 with an example of the problem under consideration. Section 4.1 discusses in detail its mathematical model adopted for consideration. The presented approach is limited to one learning and forgetting model (LFCM). Comparing the proposed approach to other models such as RC (Recency) and PID (Power Integration Diffusion) is the subject of future research. In the general case, the different shapes of the forgetting function characteristic of different professions, sex, age groups could be considered. Since there are no statistics available at this time to determine the shapes of the forgetting function for selected occupational groups, this paper uses approximate characteristics collected over the past 5 years in a population of 25-30 individuals.

Round 2

Reviewer 1 Report

Dear Authors, I think in this version the article is scientifically, methodologically and empirically on a good level. Congratulations. I wish you scientific and professional success.

Reviewer 2 Report

Accepted